# Bacterial Coinfection and Antibiotic Resistance Profiles among Hospitalised COVID-19 Patients

**DOI:** 10.3390/microorganisms10030495

**Published:** 2022-02-23

**Authors:** Abdulrahman S. Bazaid, Heba Barnawi, Husam Qanash, Ghaida Alsaif, Abdu Aldarhami, Hattan Gattan, Bandar Alharbi, Abdulaziz Alrashidi, Waleed Abu Al-Soud, Safia Moussa, Fayez Alfouzan

**Affiliations:** 1Department of Medical Laboratory Science, College of Applied Medical Sciences, University of Ha’il, Hail 55476, Saudi Arabia; h.barnawi@uoh.edu.sa (H.B.); h.qanash@uoh.edu.sa (H.Q.); g.alsaif@uoh.edu.sa (G.A.); b.alharbi@uoh.edu.sa (B.A.); am.alrashidi@uoh.edu.sa (A.A.); 2Molecular Diagnostics and Personalized Therapeutics Unit, University of Ha’il, Hail 55476, Saudi Arabia; 3Department of Medical Microbiology, Qunfudah Faculty of Medicine, Umm Al-Qura University, Al-Qunfudah 21961, Saudi Arabia; ahdarhami@uqu.edu.sa; 4Department of Medical Laboratory Technology, Faculty of Applied Medical Sciences, King Abdulaziz University, Jeddah 21589, Saudi Arabia; hsqattan@kau.edu.sa; 5Special Infectious Agents Unit, King Fahad Medical Research Center, Jeddah 22252, Saudi Arabia; 6Clinical Laboratory Sciences, Applied Medical Sciences, Jouf University, Sakaka 42421, Saudi Arabia; wabualsoud@ju.edu.sa; 7Health Sciences Research Unit, Jouf University, Sakaka 42421, Saudi Arabia; 8Department of Microbiology, King Salman Specialist Hospital, Hail 55471, Saudi Arabia; safiamoussa89@yahoo.com (S.M.); has-lab-kssh@moh.gov.sa (F.A.)

**Keywords:** Antimicrobial surveillance, COVID-19, multi-drug resistant bacteria, antibiotics, resistance

## Abstract

While it is reported that COVID-19 patients are more prone to secondary bacterial infections, which are strongly linked to the severity of complications of the disease, bacterial coinfections associated with COVID-19 are not widely studied. This work aimed to investigate the prevalence of bacterial coinfections and associated antibiotic resistance profiles among hospitalised COVID-19 patients. Age, gender, weight, bacterial identities, and antibiotic sensitivity profiles were collected retrospectively for 108 patients admitted to the intensive care unit (ICU) and non-ICU ward of a single center in Saudi Arabia. ICU patients (60%) showed a significantly higher percentage of bacterial coinfections in sputum (74%) and blood (38%) samples, compared to non-ICU. *Acinetobacter baumannii* (56%) and *Klebsiella pneumoniae* (56%) were the most prevalent bacterial species from ICU patients, presenting with full resistance to all tested antibiotics except colistin. By contrast, samples of non-ICU patients exhibited infections with *Escherichia coli* (31%) and *Pseudomonas aeruginosa* (15%) predominantly, with elevated resistance of *E. coli* to piperacillin/tazobactam and trimethoprim/sulfamethoxazole. This alarming correlation between multi-drug resistant bacterial coinfection and admission to the ICU requires more attention and precaution with prescribed antibiotics to limit the spread of resistant bacteria and improve therapeutic management.

## 1. Introduction

The COVID-19 pandemic, caused by the widespread severe acute respiratory syndrome coronavirus (SARS-CoV-2), has led to over 330 million cases and over 5 million deaths worldwide, of which 638,000 cases and 8900 deaths were recorded in the Kingdom of Saudi Arabia (KSA) [1]. Transmission of the virus occurs with direct or indirect contact with infected individuals or contaminated surfaces [2], and the infectious dose of SARS-CoV-2 is yet to be established [3]. Patients can present with asymptomatic infection or mild to very severe disease [4]. Severity of COVID-19 depends on multiple factors, including the overall immunity of the infected individuals and existing co-morbidities [4]. Patients with a compromised immune system (e.g., the elderly, neonates, HIV patients, patients with one or multiple organ failure, patients with chronic respiratory disease and other chronic diseases) are prone to suffer from severe disease [5]. Patients with the invasive form of COVID-19 typically present with very critical complications, such as sepsis, septic shock, and acute kidney injury, leading to admission to the intensive care unit (ICU) [6].

Multiple studies have reported a correlation between viral infections (e.g., the influenza virus) and bacterial pneumonia as a secondary infection (bacterial coinfection) in patients admitted to the ICU [6]. Transmission of bacterial infections within hospitals occurs by direct or indirect contact among hospitalised patients, health care workers (HCWs) and hospital equipment [7]. Ventilators and catheters are widely used during hospitalisation and are a well-known risk factor for nosocomial infections [8]. *Acinetobacter* species, *Enterobacter* species, *Enterococcus* species, *Escherichia coli*, *Klebsiella pneumoniae*, *Pseudomonas* species and *Staphylococcus* species are linked with hospital acquired infections [9]. Prolonged hospitalisation in the ICU may be linked with increased likelihood of developing bacterial coinfection in critically ill COVID-19 patients [10]. Bacterial species transmitted in hospitals are highly likely to be multi-drug resistant (MDR), which is a major challenge in managing ICU patients, leading to approximately 700,000 deaths worldwide in 2019 [11]. Co-infection and superinfection are two terms to define the detection of a second pathogen in COVID-19 patients at time of diagnosis or hospitalization, respectively [12]. About 1 in 5 of patients with SARS-CoV-2 are presented with coinfection and the majority of whom were not admitted to the ICU, while a higher percentage (41%) of superinfections was monitored among the ICU patients [13]. Nevertheless, as the differentiation between these terms is still not widely recognised, the current study will use the ‘’co-infection’’ term to describe a second infection in ICU-COVID-19 patients.

Critically ill and ICU patients are more prone to bacterial infections with increased usage of prophylactic or therapeutic antibiotics in comparison with non-invasive cases [14]. The availability of a frequently updated hospital antibiogram is therefore very important for rational prescription of antibiotics by clinicians to treat ICU patients [7]. This can ensure that effective antibiotic drugs are prescribed for emerging strains of the same bacterial species based on recommended local guidelines to ensure the success of eradicating pathogenic bacteria. In line with this initiative, the current study aimed to investigate the prevalence of bacterial coinfection among COVID-19 patients admitted to ICU and non-ICU care and to define their antibiotic sensitivity profiles.

## 2. Materials and Methods

### 2.1. Study Design

This is a retrospective cohort study conducted at King Salman hospital in Ha’il, the largest regional hospital in Ha’il designated for the admission of COVID-19 patients. Information related to COVID-19 patients, including demographics (gender, weight, and age), admission status (ICU or non-ICU) and microbiology results (bacterial co-infection and susceptibility profiles), was collected anonymously over the period of 8 months between the highest peak of the first COVID-19 wave (August 2020) and the beginning of the second wave (April 2021) in Saudi Arabia.

### 2.2. Study Cohort

Confirmed COVID-19 patients aged 18 years or over, with a positive bacterial culture of a single pathogenic bacterial species (bacterial co-infection) were included in the present study. Viral infection was confirmed with a positive polymerase chain reaction (PCR) for SARS-CoV-2. COVID-19 patients with negative bacterial growth and/or bacterial culture and those with poly-microbial growth were excluded. Mixed bacterial growth with more than one microorganism were not included to ensure a reliable and reproducible data. Samples with coagulase negative staphylococci were excluded, as it is highly expected (80%) to be a contaminant [15]. COVID-19 patients were grouped based on admission status into ICU patients and ward (non-ICU) patients.

### 2.3. Collection of Specimens for Bacterial Investigation

Various clinical specimens, including blood, urine, sputum, and wound swabs were collected from COVID-19 patients based on laboratory request made by the clinicians after physical examination and provisional diagnosis to confirm suspected bacterial coinfection. Blood was collected in BACTEC bottles and incubated in Bactec FX (Becton, Dickinson and Co., Franklin Lakes, NJ, USA) instruments for a maximum of 5 days [16]. Blood agar and MacConkey agar were used for culturing bacteria from sputum and wounds samples [17], while urine specimens were plated on sheep blood, and cystine lactose electrolyte deficient agar (CLED) and MacConkey agar plates (Oxoid, Basingstoke, UK), followed by incubation at 37 °C for 24 h. The purity of the causative bacteria was then confirmed by sub-culturing on either blood or macConkey agar plates (Oxoid, Basingstoke, UK) and incubation for 24–48 h at 37 °C. Positive bacterial cultures were further identified followed by antibiotic susceptibility testing [18].

### 2.4. Antibiotic Sensitivity Testing

Bacterial identification and antibiotic sensitivity testing were performed using a BD Phoenix™ M50 instrument (Becton, Dickinson and Co., Franklin Lakes, NJ, USA) [19]. Susceptibility of the identified Gram-negative bacteria was performed against a panel of antibiotics: amikacin, ampicillin, aztreonam, cefepime, ceftazidime, cefoxitin, colistin, ceftriaxone, cefuroxime, ciprofloxacin, ertapenem, gentamicin, imipenem, meropenem, nitrofurantoin, piperacillin, tigecycline, trimethoprim/sulfamethoxazole and piperacillin-tazobactam. Identified Gram-positive bacteria were tested against ampicillin, cephalothin, clindamycin, levofloxacin, linezolid, moxifloxacin, nitrofurantoin, penicillin, tigecycline, trimethoprim/sulfamethoxazole and vancomycin. The results of susceptibility testing were reported as ‘susceptible’ or ‘resistant’, according to Clinical and Laboratory Standards Institute guidelines CLSI document M100S-26 [20]. Quality control for several bacterial strains, including *Acinetobacter baumannii, E. faecalis*, *K. pneumoniae* and *E. coli* was required to ensure the validity of the minimum inhibitory concentration (MIC) of each antibiotic [21]. Susceptibility data were presented as percentage of resistant isolates to the total number of isolates recovered from each site or specimen for individual bacterial species.

### 2.5. Statistical Analysis

Patient demographics, bacterial identities and antibiotic resistance profiles were analysed using GraphPad Prism version 9.3.0. Differences between sample types and patient groups were assessed using unpaired *t*-test.

### 2.6. Ethics Statement

The study was approved by the Ethics Committee at Hail Affairs (reference H-08-L-074). A consent form was not required because this was retrospective study with no interaction with patients. Patient privacy and confidentiality of data were maintained in accordance with The Declaration of Helsinki.

## 3. Results

### 3.1. Demographic Characteristics of the Study Cohort

Bacterial cultures were used to investigate suspected bacterial coinfection in 108 COVID-19 patients; 65 patients (60%) were admitted to the ICU, while the remainder of the cohort (40%) were managed in non-ICU wards during the period of the study (Table 1). The percentage of males in the cohort was slightly higher (55%) than females (45%), for both ICU and non-ICU patients. Half of the study cohort were aged 56 years or over. In relation to co-morbidities, 25% of patients were diabetic and 41% were obese (BMI > 30 kg/m^2^) (Table 1).

### 3.2. Prevalence of Bacterial Coinfection in ICU and Non-ICU COVID-19 Patients

Bacterial cultures included blood, sputum, urine, and wound swabs. Sputum samples collected from both ICU and non-ICU patients represented the highest number of recovered bacteria, whereas positive cultures from wound swabs were the lowest (Figure 1). A higher number of sputum cultures from ICU patients tested positive [49/65 patients (74%)], compared with positive sputum cultures from non-ICU patients [24/43 patients (53%)] (Figure 1). In addition, the number of cases with bacteremia was higher in ICU patients [24/65 (38%)] than non-ICU patients [6/43 (16%)]. The total number of recovered bacteria from urine specimens was recorded at 8/65 (12%) and 18/43 (42%) for ICU and non-ICU patients, respectively. Statistical analysis revealed significant difference (*p* < 0.05) in the number of recovered bacteria from sputum, blood and urine from ICU and non-ICU patients.

### 3.3. Identified Bacteria from Tested Specimens among ICU and Non-ICU COVID-19 Patients

Prevalence of detected Gram-positive and Gram-negative bacterial species in specimens collected from ICU and non-ICU COVID-19 patients was investigated. Generally, *A. baumannii* was more associated with specimens collected from ICU patients, reflecting higher prevalence [37 (56%)] compared with non-ICU patients [5 (11%)]. This bacterial species was associated with blood and sputum samples. *K. pneumoniae* was detected in all specimens from ICU and non-ICU patients and was revealed to be significantly higher in blood and sputum samples from ICU (56%) compared with non-ICU (46%) patients (Figure 2). In contrast, *E. coli* was linked to specimens collected from non-ICU patients and vice versa for ICU patients, which was calculated as [14 (31%)] and [2 (3%)], respectively. *E. coli* was mainly identified in urine samples from all patients, but also in wound and sputum samples of non-ICU patients. *P. aeruginosa* was identified in both sputum and urine from 7 (15%) non-ICU patients compared to 3 (5%) ICU patients. Similar identified bacterial species were observed across all groups and specimens, with very few exclusive bacterial species, such as *Serratia marcescens* and *Serratia liquefaciens*, which were only identified in blood and sputum (Figure 2).

### 3.4. Resistance Profiles of Detected Bacterial Isolates from ICU and Non-ICU Patients

Antibiotic sensitivity patterns of the detected bacterial isolates from ICU and non-ICU COVID-19 patients were determined. Overall patterns, irrespective of the specimen, indicated that all identified bacteria from ICU patients have higher resistance to tested antibiotics compared with those from non-ICU patients (Figure 3). Resistance to all tested antibiotics was observed for *A. baumannii* identified from blood cultures of both ICU and non-ICU patients. In addition, *A. baumannii* isolated from sputum of both groups showed a high level of resistance to tested drugs except colistin. Certain *A. baumannii* strains isolated from sputum of non-ICU patients were 25% less resistant to ceftazidime, ceftriaxone, gentamicin, and trimethoprim/sulfamethoxazole compared to those from ICU patients. Isolated *K. pneumoniae* from blood and urine of ICU patients exhibited a complete resistance profile; however, detected strains in sputum from the same group showed 0% and 62% resistance to colistin and amikacin, respectively. Overall antibiotic resistance of *P. aeruginosa* isolated from ICU-patients was higher compared with non-ICU patients; however, similar percentages of resistance against ampicillin, cefoxitin, cefuroxime, ceftriaxone, cephalothin, and ertapenem were documented for the two groups. Furthermore, identified *E. coli* strains in urine specimens from ICU patients showed limited resistance profiles to ampicillin, aztreonam, cefepime, cefuroxime, ceftriaxone, cephalothin, and gentamicin. Moreover, a higher level of resistance of *E. coli* strains isolated from urine samples of non-ICU patients was recorded against piperacillin-tazobactam and trimethoprim/sulfamethoxazole in comparison with *E. coli* strains from the urine of ICU patients. Investigation of wound specimens from ICU patients failed to identify *E. coli*, while identified *E. coli* in wound swabs of non-ICU patients displayed high resistance to ampicillin, cephalothin, ciprofloxacin, levofloxacin, and trimethoprim/sulfamethoxazole.

## 4. Discussion

The ongoing COVID-19 pandemic has led to over 330 million cases and 5 million deaths worldwide up to today (23 January 2021) [22]. Admission to the ICU for at least a quarter of COVID-19 patients is highly likely because of severe respiratory complications [23]. ICU patients are more prone to secondary bacterial infections associated with prolonged ICU stay and increased risk of death [10]. This study therefore aimed to investigate the prevalence of bacterial coinfection and associated antibiotic resistance profiles among COVID-19 patients admitted to the ICU compared with non-ICU COVID-19 patients.

The data revealed a higher percentage of COVID-19 patients admitted into the ICU being diabetic (62%) or obese (65%). Obese COVID-19 patients and those with chronic illness (such as diabetes) are more likely to develop an invasive form of infection [24], and these critically ill patients typically require ICU admission [10] and mechanical ventilation [25]. The percentage of patients with bacterial coinfection in the ICU was higher (74%) than non-ICU patients (53%). This might be due to higher use of catheters, including endotracheal, arteriovenous, and urinary tubes, in ICU patients. Bacterial infection is reported to be prevalent in patients with viral respiratory infections, such influenza and COVID-19 [12,23]. COVID-19 is linked with higher levels of coinfection (12.6%) compared with influenza (8.7%), and bacterial coinfection in these patients can be used as an indicator of disease severity [26]. A recent report indicated that more than 50% of non-surviving COVID-19 patients contracted bacterial, viral and/or fungal coinfection that was associated with their deaths [27].

ICU COVID-19 patients exhibited higher rates of bacterial coinfection detected in sputum (74%) and blood (38%) compared to non-ICU patients. This finding builds on a previous claim that pulmonary secondary bacterial infections are two-fold higher in COVID-19 patients than other patients, including those with influenza [28]. Similarly, an investigation on a Chinese COVID-19 cohort showed that >50% of sputum samples collected from ICU patients tested positive in bacterial culture [29]. Bacterial pneumonia caused by multi-drug resistant organisms is highly likely to occur in severely ill COVID-19 patients, especially during mechanical ventilation [30], which is also associated with secondary bacteremia, the second most common infection among COVID-19 ICU patients [31]. COVID-19 ICU patients are reported to be five times more prone to blood infections than non-COVID-19 patients [32]. Moreover, staying in the ICU for >7 days is highly likely to expose patients to bacteremia [33]. In addition to the lower immunity of COVID-19 patients, the resulting inflammation from infection may increase permeability of the intestinal barrier and cause gut leakage, leading to possible access of bacteria into the bloodstream [31]. Furthermore, imbalances in the gut microbiota (microbiota dysbiosis) causes inflammatory dysfunction that resulted in a host been more prone to viral infection, including SARS-CoV2, and this phenomenon was reported as been well associated with obesity [34]. Besides been more susceptible to COVID-19, devolved intestinal dysbiosis in obese and/or diabetic individuals can also lead to a severe form of COVID-19 [35]. This would explain why obese and diabetics individuals in the current study are accounted for the biggest portion of the recruited COVID-19 patients.

Different bacterial species were detected in different types of samples from critically ill COVID-19 patients, particularly *A. baumannii* and *Klebsiella* species [33,36]. In the current study, Gram-negative isolates were more prevalent in all COVID-19 patients (ICU and non-ICU patients), building on previous findings that reported Gram-negative bacteria in majority of COVID-19 patients [18]. The most commonly isolated bacterial species from ICU patients in this study were *A. baumannii* and *K. pneumoniae*, especially from sputum and blood samples. *A. baumannii* has been reported to be the main pathogen in respiratory tracts of COVID-19 patients, accounting for approximately 10% of all cultured samples [37]. Another study reported that *A. baumannii* was detected in 90% of ICU COVID-19 patients [38]. Coinfection with *A. baumannii* in COVID-19 patients is significantly linked with the development of systemic infections and increased risk of motility among ICU COVID-19 patients [39,40].

*K. pneumoniae* was the most commonly isolated bacterial species from ICU and non-ICU COVID-19 patients. This bacterium has high levels of resistance to antibiotics and is able to produce various virulence factors, leading to high mortality [41]. Coinfection with this bacterium was linked with deterioration of overall health, especially in ICU COVID-19 patients [42]. *K. pneumoniae* has been reported to be the most commonly isolated bacteria from COVID-19 patients (19.4%) [43], and the current investigation showed higher prevalence of *K. pneumoniae* in non-ICU compared to ICU patients. Isolated strains of *K. pneumoniae* from blood and urine of ICU patients had complete resistance profiles; however, isolates from sputum samples for the same group presented with no resistance and 62% resistance to colistin and amikacin, respectively. This is in line with findings that the majority of *K. Pneumoniae* isolated from the blood of ICU patients were multi-drug resistant with high resistance to most used antibiotics [44].

*E. coli* was the third most commonly detected bacterial species (9.7%) in both groups of patients, while being the second in non-ICU COVID-19 patients (24.6%), isolated mainly from urine. Similarly, *E. coli* was previously identified in 16% of COVID-19 patients [45]. Strains of *E. coli* isolated from non-ICU patients displayed higher resistance to ampicillin, cephalothin, ciprofloxacin, levofloxacin, and trimethoprim/sulfamethoxazole. The observed overall high resistance confirmed previous findings that most uropathogens *E. coli* in Hail were resistant to the majority of antibiotics, including trimethoprim/sulfamethoxazole and piperacillin [46]. However, prevalence and antibiotics resistance profiles of bacterial coinfection may not be the same across isolates from different hospitals/locations.

Varying prevalence and resistance profiles of bacterial coinfection among COVID-19 patients have been reported. In Bahrain, the prevalence of the most common Gram-negative isolates from over 1380 COVID-19 patients was as follows: *K. pneumoniae* (23.8%), *P. aeruginosa* (23.2%), *A. baumannii* (22.0%), and *E. coli* (17.1%) [47]. In addition, among 1495 COVID-19 patients hospitalized in Wuhan with secondary bacterial infections, the most common isolated bacteria were *A. baumannii* (35.8%) and *K. pneumoniae* (30.8%) [48]. These findings differ from the data collected in the current study for ICU and non-ICU COVID-19 patients. The observed prevalence of bacterial coinfection among ICU COVID-19 patients might be attributed to several factors including compromised immunity, poor adherence to self-protective measures by healthcare workers and patients, low standard of infection control between wards, high workload, and staff shortage [37]. Therefore, it is important to monitor bacterial coinfection in COVID-19 patients, especially with multi-drug resistant bacteria, to control infections in hospitals.

Different resistance patterns of bacterial coinfection against commonly used antibiotics were determined. *A. baumannii* strains recovered from sputum of ICU COVID-19 patients were more resistant against ceftazidime, ceftriaxone, gentamicin, and trimethoprim/sulfamethoxazole compared to isolates from non-ICU patients. This observation and the emergence of completely resistant strains in ICU specimens can be attributed to overuse of antibiotics in the ICU during the pandemic, leading to an extensive selection pressure for resistance and increased dissemination of multi-drug resistant organisms [49]. Moreover, strains of *A. baumannii* isolated from the blood of both COVID-19 groups showed resistance to all tested antibiotics. Strains isolated from sputum samples from ICU patients were sensitive to colistin. This observation is in line with a previous study conducted in the same region (Hail), which reported resistance of *A. baumannii* from COVID-19 and non-COVID-19 individuals to all antibiotics except colistin (5% resistance rate) and ertapenem [50]. Another study showed that *A. baumannii* isolated from endotracheal aspirate samples of nineteen COVID-19 patients was resistant to all tested antibiotics, except colistin with a resistance rate of 52% [36]. Furthermore, isolated *A. baumannii* from blood among Greek ICU COVID-19 patients was resistant (100%) to meropenem and amikacin (87.5%), but more sensitive to colistin (47%) and tigecycline (27.5%) [51]. These surveillance studies indicate that colistin is an active drug against *A*. *baumannii* strains and can be used to treat secondary bacterial pneumonia in COVID-19 patients caused by this bacterium, although completely resistant strains were also recorded in the current investigation. Completely resistant *A. baumannii* strains isolated from the blood of COVID-19 patients require urgent attention to identify combination therapy active towards this emerging resistance.

Restricted access to the patients’ files resulted in several limitations, including unavailable information regarding the length of hospitalisation and the outcome of COVID-19 management after diagnosis of bacterial coinfection and prescription of antibiotics. This information is useful to the analysis of possible correlations between bacterial resistance patterns, prescribed antibiotics, and the outcome of treatment.

## 5. Conclusions

COVID-19 patients admitted to ICU had higher bacterial coinfection compared to non-ICU patients, with the majority of bacterial strains isolated from sputum and blood being Gram-negative bacteria, including *A. baumannii* and *K. pneumoniae*. *A. baumannii,* were resistant to all tested antibiotics except colistin, which can be used in treatment plans. A high level of resistance among Gram-negative bacteria identified from COVID-19 patients was observed, especially in ICU patients. The findings of the current study suggest that continuous monitoring of bacterial coinfection and resistance patterns, as well as improving infection control measures, are important to control the pandemic at a local and global level.

## Figures and Tables

**Figure 1 microorganisms-10-00495-f001:**
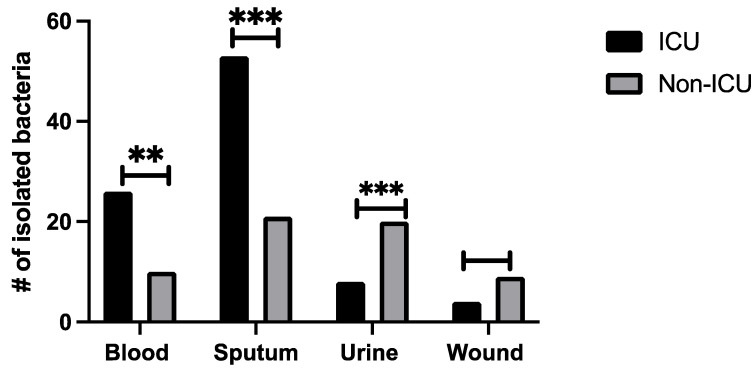
The numbers of bacterial isolates identified in different types of samples collected from ICU and non-ICU COVID-19 patients; ** *p* < 0.01, *** *p* < 0.001.

**Figure 2 microorganisms-10-00495-f002:**
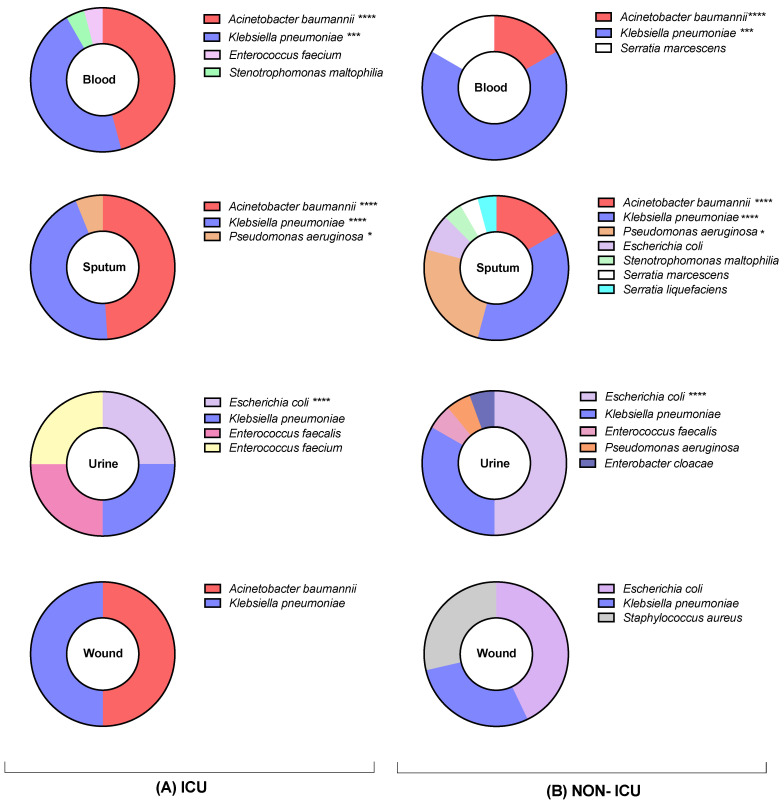
The numbers of identified bacterial species isolated from COVID-19 patients admitted to (**A**) ICU and (**B**) non-ICU in different sample types (blood, sputum, urine, and wound swabs); * *p* < 0.05, *** *p* < 0.001, **** *p* < 0.0001.

**Figure 3 microorganisms-10-00495-f003:**
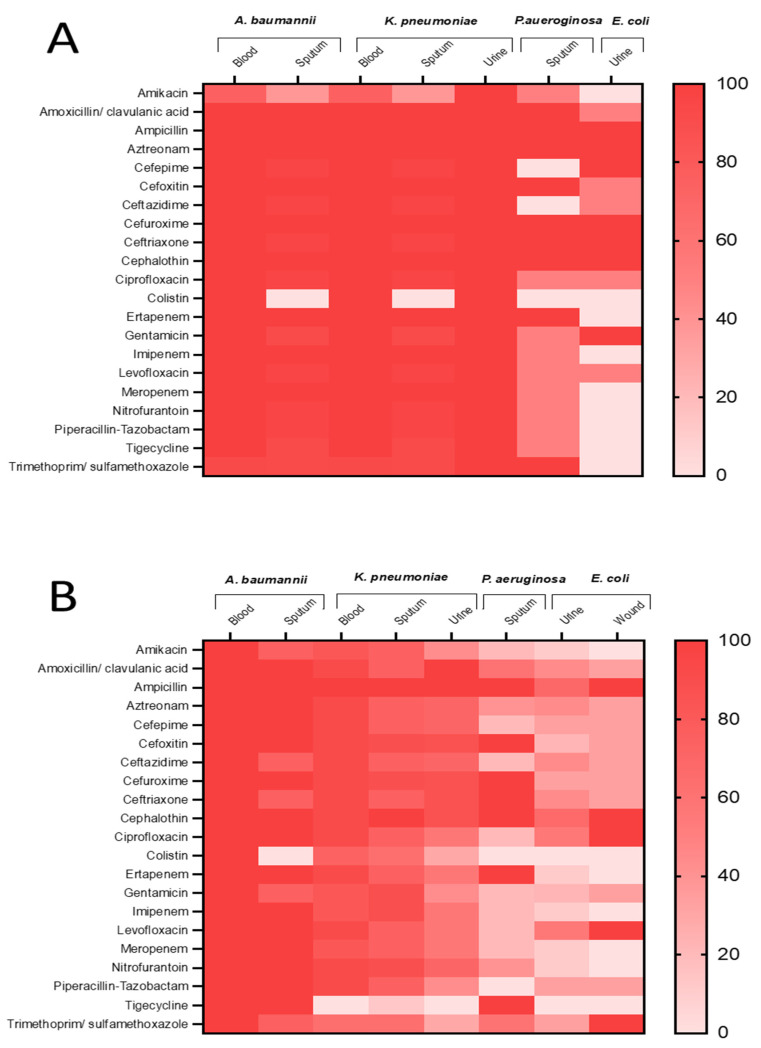
Heat maps representing the percentages of antibiotic resistant bacteria in positive cultures isolated from samples collected from COVID-19 patients in (**A**) ICU and (**B**) non-ICU care.

**Table 1 microorganisms-10-00495-t001:** Demographic and clinical characteristics of COVID-19 patients admitted to ICU and wards (non-ICU patients).

	ICU Patients (%)	Non-ICU Patients (%)	Total
**Gender**			
Male	37 (62)	22 (38)	59
Female	28 (56)	21 (44)	49
**Age**			
25–34	-	3 (100)	3
35–44	4 (50)	4 (50)	8
45–54	9 (56)	7 (44)	16
55–64	13 (52)	12 (48)	25
65+	39 (70)	17 (30)	56
**BMI**			
Underweight (>18.5)	-	-	
Normal (18.8–29.5)	36 (56)	28 (43)	64
Obese (<30)	29 (66)	15 (34)	44
**Underlining disease**			
Diabetic	17 (63)	10 (37)	27
Non-diabetic	48 (59)	33 (41)	81

Data are presented in numbers and percentages in parentheses (%).

## Data Availability

All data are available within the manuscript.

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
