# Peer review of "Bacterial Coinfection and Antibiotic Resistance Profiles among Hospitalised COVID-19 Patients"

_microorganisms, 2022, doi:10.3390/microorganisms10030495_

Round 1

Reviewer 1 Report

Dear Authors,

After the review process, I have several comments: you should include more numerical data in the abstract; you should include more comments, in the discussions section) about the patients related to the link between microbiota fingerprint and patients pathologies (e.g., the link between obesity and microbiota dysbiosis); you should include references in all Materials and Methods sections; the role of microbiota is essential here and the presentation of alternative data are important to increase the value of the paper. Best regards.

Author Response

Dear Reviewer,

We would like to thank you for the comments that were helpful and we appreciate the efforts taken to improve our manuscript. The response to each point raised has been detailed below.

LIST OF COMMENTS AND RESPONSES:

REVIEWER 1

____________________________________

COMMENT (1): you should include more numerical data in the abstract;

RESPONSE (1): This was addressed.

____________________________________

COMMENT (2): you should include more comments, in the discussions section) about the patients related to the link between microbiota fingerprint and patients pathologies (e.g., the link between obesity and microbiota dysbiosis); the role of microbiota is essential here and the presentation of alternative data are important to increase the value of the paper.

RESPONSE (2): This was addressed. Although the scope of the study is to assess the prevalence of coinfections and antibiotic susceptibility among ICU and non-ICU COVID-19 patients, this section was added to the discussion as we believe it was a very valuable comment.

____________________________________

COMMENT (3): you should include references in all Materials and Methods sections.

RESPONSE (3): This was addressed. 

Reviewer 2 Report

This manuscript by Abdulrahman S. Bazaid et al. describes the data on bacterial coinfection in COVID-positive patients.

This manuscript needs to be thoroughly revised before possible acceptance.

Global:

Italics: "e.g.," bacterial names.

Introduction: 

Authors should detail the epidemiology of superinfection versus coinfection.

Methods: 

Line 74: the largest regional hospital in Saudi Arabia...or a particular region.

Authors should justify their inclusion process: inclusion period; number of patients to be included; but also exclusion criteria (why exclude patients with polymicrobial growth?)

line 91: what are "appropriate agars"? Provide details.

Clarify the version of the CLSI guidelines 

Ethics statement: clarify whether patients were given appropriate information and consent.

Results:

Why did the authors not consider respiratory conditions as underlying diseases?

For positive blood cultures with positive contaminant (CNStaph), what is the authors' consideration?

Figure 3: Please consider reversing the figure, producing 3 figures (blood; sputum; and urine) to compare COVID-positive and -negative patients. Would this approach be more appropriate for the purpose of the study?

Discussion: 

Line 193: specify date.

Conclusion: 

Line 305 is not especially supported by this study. Please amend the conclusion.

Author Response

Dear Reviewer,

We would like to thank you for the comments that were helpful and we appreciate the efforts taken to improve our manuscript. The response to each point raised has been detailed below.

____________________________________

COMMENT (1):  Global:

RESPONSE (1): Although it is unclear the section associated with this comment, We added it to conclusion.

____________________________________

COMMENT (2): Italics: "e.g.," bacterial names.

RESPONSE (2): This was addressed.

____________________________________

COMMENT (3): Introduction: Authors should detail the epidemiology of superinfection versus coinfection.

RESPONSE (3): We appreciate this comment, but the differentiation between coinfection and superinfection terms might still emerging and not yet well recognised since the majority of cited references used the terms ‘’coinfection’’. We have decided to add few lines about these two terms with overall epidemiology as requested.

____________________________________

COMMENT (4): Methods: Line 74: the largest regional hospital in Saudi Arabia...or a particular region.

RESPONSE (4): This was addressed. ____________________________________

COMMENT (5): Authors should justify their inclusion process: inclusion period; number of patients to be included; but also exclusion criteria (why exclude patients with polymicrobial growth?)

RESPONSE (5): This information was added. Patient with polymicrobial bacterial growth were excluded because it makes interpretation of causative bacteria more difficult and prone to error and .

____________________________________

__________________________________

COMMENT (6): line 91: what are "appropriate agars"? Provide details.

RESPONSE (6): This was addressed.

COMMENT (7): Clarify the version of the CLSI guidelines

RESPONSE (7): This was addressed.

____________________________________

COMMENT (8): Ethics statement: clarify whether patients were given appropriate information and consent.

RESPONSE (8): This was added. No consent form was given because this is a retrospective study and there was not any kind of interaction with the patient at all. 

 ____________________________________

COMMENT (9): Results: Why did the authors not consider respiratory conditions as underlying diseases?

RESPONSE (9): Signs and symptoms of patients were not available due to limited access to patient information.  In addition, all patients included in this study were COVID-19 positive based on PCR testing regardless of the symptoms. This is because symptoms of COVID19 is overlapping with other respiratory condition, and there was not any reported respiratory condition in these patients accept from COVID-19 based on the PCR testing.  Thus, we apologies as will not be able to conduct this suggested comment. 

____________________________________

COMMENT (10): For positive blood cultures with positive contaminant (CNStaph), what is the authors' consideration?

RESPONSE (10): This section was added to Discussion. 

____________________________________

____________________________________

COMMENT (11): Figure 3: Please consider reversing the figure, producing 3 figures (blood; sputum; and urine) to compare COVID-positive and -negative patients. Would this approach be more appropriate for the purpose of the study?

RESPONSE (11):  We appreciate this comment, but all obtained data in this study are belong to patients with positive COVID-19 (based on the PCR test), and there was not any COVID-19 negative to do this suggested comparison. Also, the aim of the study is to compare the prevalence of bacterial coinfection and antibiotic sensitivity profiles among COVDI-19 patients in ICU and non-ICU.  

____________________________________

COMMENT (12): Discussion: Line 193: specify date.

RESPONSE (12): This was addressed.

____________________________________

COMMENT (13): Conclusion: Line 305 is not especially supported by this study. Please amend the conclusion.

RESPONSE (13): This was addressed.

____________________________________

Round 2

Reviewer 1 Report

Dear Authors,

No other comments compared to the first round of review. Best regards.

Author Response

--> Comment: No other comments compared to the first round of review. Best regards.

--> Response: We appreciate your help and support to improve our manuscript. 

Reviewer 2 Report

The manuscript has been revised. Nevertheless, some of the precedent issues remain.

COMMENT (5): Authors should justify their inclusion process: inclusion period; number of patients to be included; but also exclusion criteria (why exclude patients with polymicrobial growth?)

RESPONSE (5): This information was added. Patient with polymicrobial bacterial growth were excluded because it makes interpretation of causative bacteria more difficult and prone to error and .

--> Comment : Thanks to dilution thresholds, this possible bias could be easily overcome. Please consider and discuss.

COMMENT (8): Ethics statement: clarify whether patients were given appropriate information and consent.

RESPONSE (8): This was added. No consent form was given because this is a retrospective study and there was not any kind of interaction with the patient at all.

--> Comment : As clinical information are described in the present study, the authors have to ensure that they, at least, inform included patients or their family.

COMMENT (10): For positive blood cultures with positive contaminant (CNStaph), what is the authors' consideration?

RESPONSE (10): This section was added to Discussion.

--> Comment : Discussion is not sufficient. Authors have to ensure themselves that they did not consider potential contaminant as clinically relevant (double or triple positive sampling).

COMMENT (11): Figure 3: Please consider reversing the figure, producing 3 figures (blood; sputum; and urine) to compare COVID-positive and -negative patients. Would this approach be more appropriate for the purpose of the study?

RESPONSE (11): We appreciate this comment, but all obtained data in this study are belong to patients with positive COVID-19 (based on the PCR test), and there was not any COVID-19 negative to do this suggested comparison. Also, the aim of the study is to compare the prevalence of bacterial coinfection and antibiotic sensitivity profiles among COVDI-19 patients in ICU and non-ICU.

--> Comment : please excuse my previous comment. My preivous comment should be consider with ICU or non-ICU patients and not COVID or non-COVID patient.

Author Response

Dear Reviewer 

COMMENT (5): Authors should justify their inclusion process: inclusion period; number of patients to be included; but also exclusion criteria (why exclude patients with polymicrobial growth?)

RESPONSE (5): This information was added. Patient with polymicrobial bacterial growth were excluded because it makes interpretation of causative bacteria more difficult and prone to error.

--> Comment: Thanks to dilution thresholds, this possible bias could be easily overcome. Please consider and discuss.

--> Response: We totally agree that polymicrobial cultures can easily be identified and further analysed. However, according to the local policy of the hospital, the obtained data from any ‘’polymicrobial culture’’ must be withdrawn and not further analysed, while a new specimen (e.g swab, urine , blood ect) is collected from patients for a second round of testing. Thus, unavailable identity and sensitivity testing of all polymicrobial cultures has forced investigators to exclude them from the study.

COMMENT (8): Ethics statement: clarify whether patients were given appropriate information and consent.

RESPONSE (8): This was added. No consent form was given because this is a retrospective study and there was not any kind of interaction with the patient at all.

--> Comment : As clinical information are described in the present study, the authors have to ensure that they, at least, inform included patients or their family.

--> Response: We appreciate this comment. However, we should have mentioned that all data were anonymously collected without names or contact details of patients (This information was added). Also, this study was only conducted after an appropriate ethical approval was granted and a consent form was waived by the committee.  

COMMENT (10): For positive blood cultures with positive contaminant (CNStaph), what is the authors' consideration?

RESPONSE (10): This section was added to Discussion.

--> Comment : Discussion is not sufficient. Authors have to ensure themselves that they did not consider potential contaminant as clinically relevant (double or triple positive sampling).

--> Response: This is very helpful comment. We totally agree that blood cultures with identified CoNs from blood cultures are potentially contaminant and should not be included in the study to ensure high level of reliability. We also agree that more sampling should be done, but the local hospital do not follow this path.  They rather  treat identified CoNs in immunosuppressed patients like (ICU patients) as positive and elsewhere is negative. Therefore, we have decided to exclude any identified CoNs from this study to ensure that all date are not prone to possible contamination (tis was added to methods).

COMMENT (11): Figure 3: Please consider reversing the figure, producing 3 figures (blood; sputum; and urine) to compare COVID-positive and -negative patients. Would this approach be more appropriate for the purpose of the study?

RESPONSE (11): We appreciate this comment, but all obtained data in this study are belong to patients with positive COVID-19 (based on the PCR test), and there was not any COVID-19 negative to do this suggested comparison. Also, the aim of the study is to compare the prevalence of bacterial coinfection and antibiotic sensitivity profiles among COVDI-19 patients in ICU and non-ICU.

--> Comment : please excuse my previous comment. My preivous comment should be consider with ICU or non-ICU patients and not COVID or non-COVID patient.

--> Response: Actually, we do appreciate the way you suggested to present this data. However, if possible, we kindly prefer to stick on the current figure, which shows an overview data without focusing on each source of specimens (blood; sputum; and urine) as we believe that this presentation might be well matched with the aim of the study.